# Significance of Micromorphological Characteristics and Expression of Intermediate Filament Proteins CK7 and CK20 in the Differential Diagnosis of Serrated Lesions of the Colorectum

Ivan Ilić [1], Pavle Ranđelović [2,*], Žaklina Mijović [1], Maja Jovičić Milentijević [1], Biljana Radovanović Dinić [3] and Jana Cvetković [4]

1  Center for Pathology and Pathological Anatomy, University Clinical Center Niš, Faculty of Medicine, University of Niš, 18000 Niš, Serbia
2  Department of Physiology, Faculty of Medicine, University of Niš, 18000 Niš, Serbia
3  Clinic for Gastroenterology and Hepatology, University Clinical Center Niš, Faculty of Medicine, University of Niš, 18000 Niš, Serbia
4  Department for Pathology, General Hospital Leskovac, 16000 Leskovac, Serbia
*  Correspondence: pavleus@gmail.com

**Abstract:** Serrated lesions in the colorectum include all epithelial neoplastic lesions, which show a sawtooth-like morphology in the epithelial crypts. Classification systems nosologically divide colon serrated polyps into three different categories, primarily emphasizing their micromorphological growth pattern and cytodifferentiation: (1) hyperplastic polyps, (2) sessile serrated adenomas/polyps and (3) traditional serrated adenomas. Overall, 109 patients with serrated lesions of the colon, who underwent endoscopic or surgical polypectomy/tumorectomy during one or multiple endoscopic or surgical interventions, over a four-year period, were analyzed. The average age of patients was $62.8 \pm 11.6$ years. The frequency of serrated lesions of the colon in male patients was 2.4 times higher than in females (70.6% vs. 29.4%). All sessile serrated lesions without dysplasia were positive for CK7 and statistically significant compared to other serrated lesions, if this positivity was present in the complete crypt ($p = 0.005$). CK20 positivity, which is limited to the upper half of the crypt, is a special feature of hyperplastic polyps compared to other serrated lesions, which is statistically significant ($p = 0.0078$). Whereas, CK20 positivity of complete crypts is a statistically significant feature of traditional serrated adenomas ($p < 0.01$). Differences in the expression pattern of cytokeratin 7 and 20 in different serrated lesions may indicate different pathways of colorectal carcinogenesis, and be diagnostically and prognostically useful.

**Keywords:** serrated; classification; lesion; colon; rectum

## 1. Introduction

A review of the literature in the last three decades reveals several changes in the classification of serrated lesions of the large intestine, which primarily highlight their histological patterns and cytodifferentiation. However, in practical use, these changes can introduce a certain degree of subjectivity, with some subtypes of serrated lesions being almost anecdotal in nature. Thanks to a better understanding of the pathology of these lesions and a better histopathological analysis, the World Health Organization issued a new classification of serrated lesions in 2019, which is based on the morphological components of the lesions, the level of dysplasia and the molecular pathology of the serrated lesion [1]. Serrated polyps are a heterogeneous group of lesions characterized by a serrated morphology of the epithelium (star-shaped or saw-tooth-like), which includes (1) hyperplastic polyps (HP) of the microvesicular type (MVHP) and goblet cell rich subtype (goblet cell rich-GCHP), while the mucin-poor subtype is not recognized in the latest classification; (2) sessile serrated lesions (SSL) (without and with dysplasia); (3) traditional serrated adenomas (TSA); (4) unclassified serrated adenomas (NSA).

HPs consist of superficial serrated epithelium and funnel-shaped, evenly spaced crypts with proliferative zones confined to the bases of the crypts. Crypts do not show basal dilatation, significant distortion or submucosal displacement, but branching of individual crypts may occur. Serration of the epithelium in HP can be seen in the surface epithelium and the superficial part of the crypts [2].

The distinction of SSL from HP (especially MVHP) and TSA is based mainly on architecture, although cytological features also play an important role [2]. Changes in crypt architecture in SSL are defined as horizontal growth along the muscular sheet of the mucosa, widening of the crypt base (basal third of the crypt), serrations extending into the bases of the crypts and asymmetric proliferation [2,3].

The two most distinctive features of TSA are fissure-like serrations (resembling the narrow slits normally seen in the mucosa of the small intestine) and tall cylindrical cells with intensely eosinophilic cytoplasm, with striated nuclei. Ectopic crypts are formations defined as epithelial buds that are not attached to the muscle sheet of the mucous membrane and are always located along the sides of villous projections of protuberant TSAs, but are rarely present (and not necessary for a diagnosis) in flat TSAs [1]. Some dysplastic polyps with serrated morphology are difficult to classify as TSA or SSL with dysplasia, especially if they are superficial biopsy specimens [4].

Several studies showed the phenotypic expression of CK7 and CK20 in certain lesions, where during the process of carcinogenesis some dysplastic glands were CK7 positive, and the normal mucosa was CK7−/CK20+ [5,6]. In colorectal cancers, the superficial parts of the glands of the normal mucosa express CK20 and remain negative for CK7, while the crypts may show CK7 positivity [7,8]. Other authors reported that the CK7−/CK20+ immunophenotype is a significant immunohistochemical feature of colorectal adenocarcinomas [9].

Both markers can have different patterns of expression in tumor cells in the sense that they are diffusely positive for CK20 or that the expression is significantly reduced [10], and some earlier studies reported that tumor cells can express CK7 from 0% to 74% [7,11,12]. Most serrated lesions and hyperplastic polyps (HP) are incidental findings discovered in appendices that have been removed for other reasons [13].

Considering the limited knowledge about the biological behavior of serrated lesions, more efforts are needed to detect and collect cases with adequate clinical follow-up, which could be subjected to studies on their malignant potential [14].

## 2. Materials and Methods

The four most important diagnostic criteria for SSL were (1) strong serrated morphology of the lower third of the crypt (with or without branching); (2) T and L shapes of crypts above muscular layer of mucosa; (3) inverted crypts (pseudoinvasion) below muscular layer of mucosa; and (4) cylindrical dilatation of lower third of crypts (with or without mucins). Having in mind that, to date, there are still no quantitative criteria for the diagnosis of SSL, due to pragmatic reasons, we used the presence of two out of four diagnostic determinants in at least two different crypts to make a diagnosis. According to microscopic criteria for the diagnosis of serrated lesions, the analysis included all serrated lesions diagnosed in the mentioned time frame, while the exclusion criteria were hyperplastic polyps occurring synchronously with conventional adenomas and carcinomas which were not of the serrated type.

Overall, 109 cases of serrated colon lesions of all histopathological subtypes, in the same number of patients, diagnosed in the period from 2017 to 2020, were analyzed. The study included 34 cases of hyperplastic polyps, 27 cases of sessile serrated lesions without dysplasia, 11 cases of sessile serrated lesions with dysplasia, 6 cases of traditional serrated adenomas, 21 cases of mixed serrated lesions, 2 unclassified serrated lesions and 8 malignantly altered serrated lesions. In this study, tissue samples of serrated lesions were used, obtained as colonoscopic biopsies or polypectomy samples from the Gastroenterology and Hepatology Clinic of the Niš University Hospital, or obtained as part of partial and

complete colectomies in patients surgically treated at the University Medical Center in Niš. All samples have already been routinely processed to paraffin molds.

For the purposes of immunohistochemical staining, the most characteristic microscopic sample (region) was taken from the lesion area and subjected to this specific procedure in the form of a paraffin mold. According to the basic morphological assessment made on HE preparations, one mold was selected that contained a tissue sample of the serrated lesion in its entirety, or from an average representative region of serrated lesions of larger diameters. Microtome sections with a thickness of about 4 μm were made from paraffin molds, adhered to "super frost" slides and stained immunohistochemically for CK7 (Monoclonal Mouse Anti-Human Cytokeratin 7 Antigen; Ready-to-use; Clone OV-TL 12/30; Code IR619) and CK20 (Monoclonal Mouse Anti-Human Cytokeratin 20 Antigen; Ready-to-use; Clone Ks20.8; Code IR777). After deparaffinization and hydration of the sections using xylene and a series of alcohols of decreasing concentration, antigen unmasking was performed in a microwave oven for 20 min in a citrate buffer, followed by cooling at room temperature, washing and blocking of endogenous peroxidase with 3% hydrogen peroxide. The sections were washed in PBS (Phosphate Buffered Saline), pH = 7.4, and then the primary antibody was applied followed by a 40min incubation at room temperature. Washing with PBS was then followed by Labeled Streptavidin-Biotin 2 System, Horseradish Peroxidase (LSAB2 System-HRP, 15 mL, Code K0673); all antibodies used originated from the same manufacturer (DAKO, Glostrup Denmark), and the visualization system used was En Vision with chromogenic DAB (DAKO Cytomation). Background staining was performed with Mayer's hematoxylin (Merck, Germany) with a standard exposure time, and other standardized procedural aids, and finally dehydrated and illuminated (short 75% ethanol, short 96% ethanol, 5 min in absolute alcohol and 10 min in xylene) and covered with DPX and a cover slip.

According to the recommendations of some authors, the expressions of CK7 and CK20 were assessed by staining the cytoplasmic membrane and cytoplasm, and if less than 5% of cells expressed these cytokeratins, in any part of the crypt, the staining was interpreted as negative [8].

All statistical calculations were performed using Excel 2013 and Jandel Sigma Stat 2.0 (SPSS Inc., Chicago, IL, USA). The distribution of normality for parametric data was determined by the Shapiro–Wilk Normality test. The results for monitored parameters were analyzed using descriptive statistics (average and standard deviation), comparative tests (parametric (*t*-test for normal distribution of data) and non-parametric (Fisher test) and correlation tests (Pearson test of linear correlation for normal distribution). The assessment of the significance of the determined differences in morphological and clinical parameters was considered significant for a value of $p < 0.05$, and for immunohistochemical parameters, for a value of $p < 0.01$.

## 3. Results

The average age of patients was $62.8 \pm 11.6$ years, where the youngest patient was 21 years old and the oldest was 86 years old. Serrated lesions of the colon are divided into 7 categories: (1) hyperplastic polyp, (2) sessile serrated lesion without dysplasia, (3) sessile serrated lesion with dysplasia, (4) traditional serrated adenoma, (5) mixed serrated lesion, (6) unclassified serrated lesion and (7) malignantly altered serrated lesion (Figure 1). The largest number of patients, or 34 (31.2%), had hyperplastic polyp(s), while sessile serrated lesion without dysplasia, sessile serrated lesion with dysplasia, traditional serrated adenoma, mixed serrated lesion, unclassified serrated lesion and malignantly altered serrated lesion were diagnosed in 27 (24.8%), 11 (10.1%), 6 (5.5%), 21 (19.3%), 2 (1.8%) and 8 (7.3%) patients, respectively.

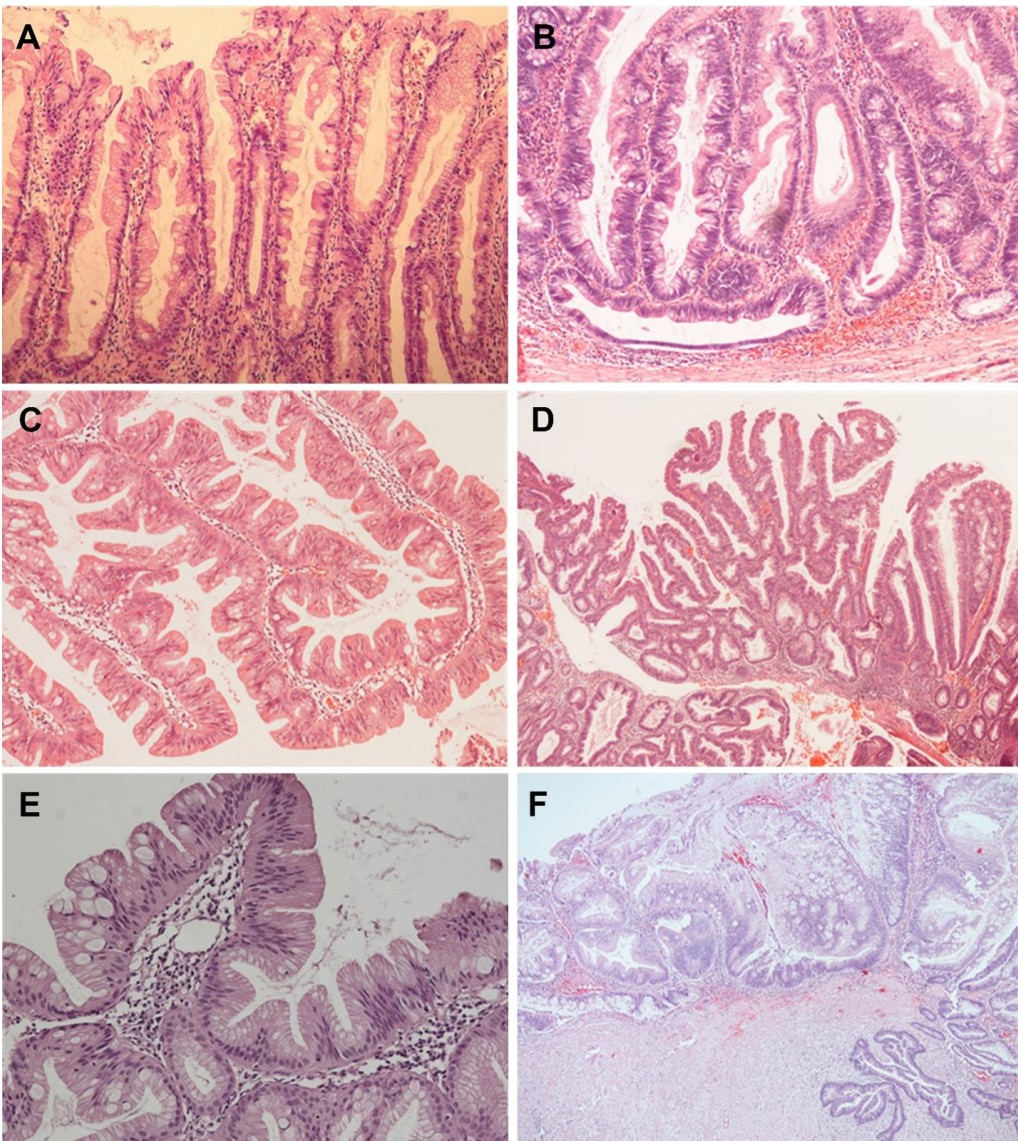

**Figure 1.** (**A**) Characteristic serration of the upper part of the crypt epithelium in hyperplastic polyp (HE × 100); (**B**) Horizontal growth of the crypt base in the form of an inverted T shape in a SSL with dysplasia (HE × 100); (**C**) The surface of a TSA with a villiform aspect with pronounced serrations and numerous ectopic crypt formations (HE × 100); (**D**) Mixed serrated lesion composed of TSA and SSL elements with dysplasia (HE × 40); (**E**) Superficial parts of an unclassified serrated lesion that can be interpreted as a serrated tubulovillous adenoma on biopsy (HE × 200); (**F**) SSL of the appendix (left) with dysplasia as a precursor of serrated adenocarcinoma (right) (HE × 40).

The frequency of serrated lesions of the colon in male patients was 2.4 times higher than in females (70.6% vs. 29.4%). The differences in the average ages of patients with different categories of serrated colon lesions were not statistically significant (*t*-test, *p* > 0.05).

In relation to the localization and type of the serrated lesion, sessile serrated lesions without dysplasia were statistically significantly more often if diagnosed in the localizations of the ascending and sigmoid parts of the colon compared to other serrated colonic lesions of the same localization (*p* < 0.05). In addition, a malignant alteration of serrated lesions occurred statistically significantly more often in the anal canal compared to other locations (*p* = 0.0254) (Table 1).

**Table 1.** Relationship between the histological type of serrated lesions and their localization in the large intestine.

| Microscopic Appearance/Localization | Appendix | Caecum | Ascending Colon | Transverse Colon | Descending Colon | Sigmoid Colon | Rectum | Anal Canal | ∑ (n) |
|---|---|---|---|---|---|---|---|---|---|
| HP | 5 | 2 | 1 | 3 | 5 | 11 | 7 | 0 | 34 |
| SSL without dysplasia | 4 | 2 | 5 * | 1 | 1 | 4 * | 8 | 2 | 27 |
| SSL with dysplasia | 0 | 1 | 0 | 2 | 1 | 4 | 3 | 0 | 11 |
| TSA | 0 | 0 | 0 | 0 | 0 | 3 | 3 | 0 | 6 |
| Mixed serrated lesion | 1 | 0 | 3 | 1 | 2 | 8 | 6 | 0 | 21 |
| Unclassified serrated lesion | 0 | 0 | 0 | 1 | 0 | 0 | 1 | 0 | 2 |
| Malignantly altered serrated lesion | 1 | 1 | 0 | 1 | 0 | 1 | 2 | 2 * | 8 |
| ∑ (n) | 11 | 6 | 4 | 9 | 9 | 27 | 30 | 2 | 109 |

* $p < 0.05$.

The analysis of the macroscopic manner of growth of serrated lesions of the large intestine singled out sessile serrated lesions with dysplasia and traditional serrated adenomas as lesions that are statistically significantly more often present as semi-pedunculated, i.e., semi-sessile, compared to other serrated lesions ($p < 0.05$).

The largest macroscopic diameter of all serrated lesions was 40 mm in the category of malignantly altered serrated lesions, which, at the same time, statistically significantly more often had an average larger diameter compared to other serrated lesions (*t*-test, $p < 0.05$). The smallest macroscopically recorded diameter of 2 mm was present in hyperplastic polyps, sessile serrated lesions with dysplasia and mixed serrated lesions, but the average smallest diameter of serrated lesions was 5.5 mm in the category of hyperplastic polyps, which is statistically significantly more common compared to other serrated lesions. (*t*-test, $p < 0.05$). A weak positive correlation was found between the diameter of the serrated lesions and the age of patients, which was not statistically significant (r = 0.165; $p = 0.558$).

The analyzed serrated lesions of the large intestine did not show immunoreactivity to CK7 in 28.4% (31 cases), which was statistically significant for all TSAs (Fisher's test, $p = 0.00036$) compared to other serrated lesions, as well as for malignantly altered serrated lesions in 75% of the cases ($p = 0.0058$). Considering that all TSAs were negative (Figure 2A), CK7 immunoreactivity was observed only in cases of mixed serrated lesions, where TSAs were one of the components (Figure 2B).

All sessile serrated lesions without dysplasia were positive for CK7 and statistically significant compared to other serrated lesions if such positivity was present in the complete crypt ($p = 0.005$) (Figure 2C). CK7 positivity in the upper half of the crypts shows that hyperplastic polyps are statistically significantly more often immunoreactive in the upper half of the crypts compared to other serrated lesions (Figure 2D). Given that malignantly altered serrated lesions are significantly negative in 75% of the cases, the positivity encountered in 25% of the cases is statistically significantly limited to the upper half of the crypts ($p = 0.0019$) (Table 2).

Hyperplastic polyps showed immunoreactivity along the entire crypts in only two cases, which was not statistically significant ($p > 0.01$) (Figure 2E).

CK20 analysis showed that positivity was present in 98.2% of cases. Negative cases were present only in the category of unclassified serrated lesions (1.8%). CK20 positivity, which is limited to the upper half of the crypt, is a special feature of hyperplastic polyps compared to other serrated lesions, and is statistically significant ($p = 0.0078$) (Figure 2F), while CK20 positivity of complete crypts is a statistically significant feature of traditional serrated adenomas ($p < 0.01$) (Figure 2G) (Table 3).

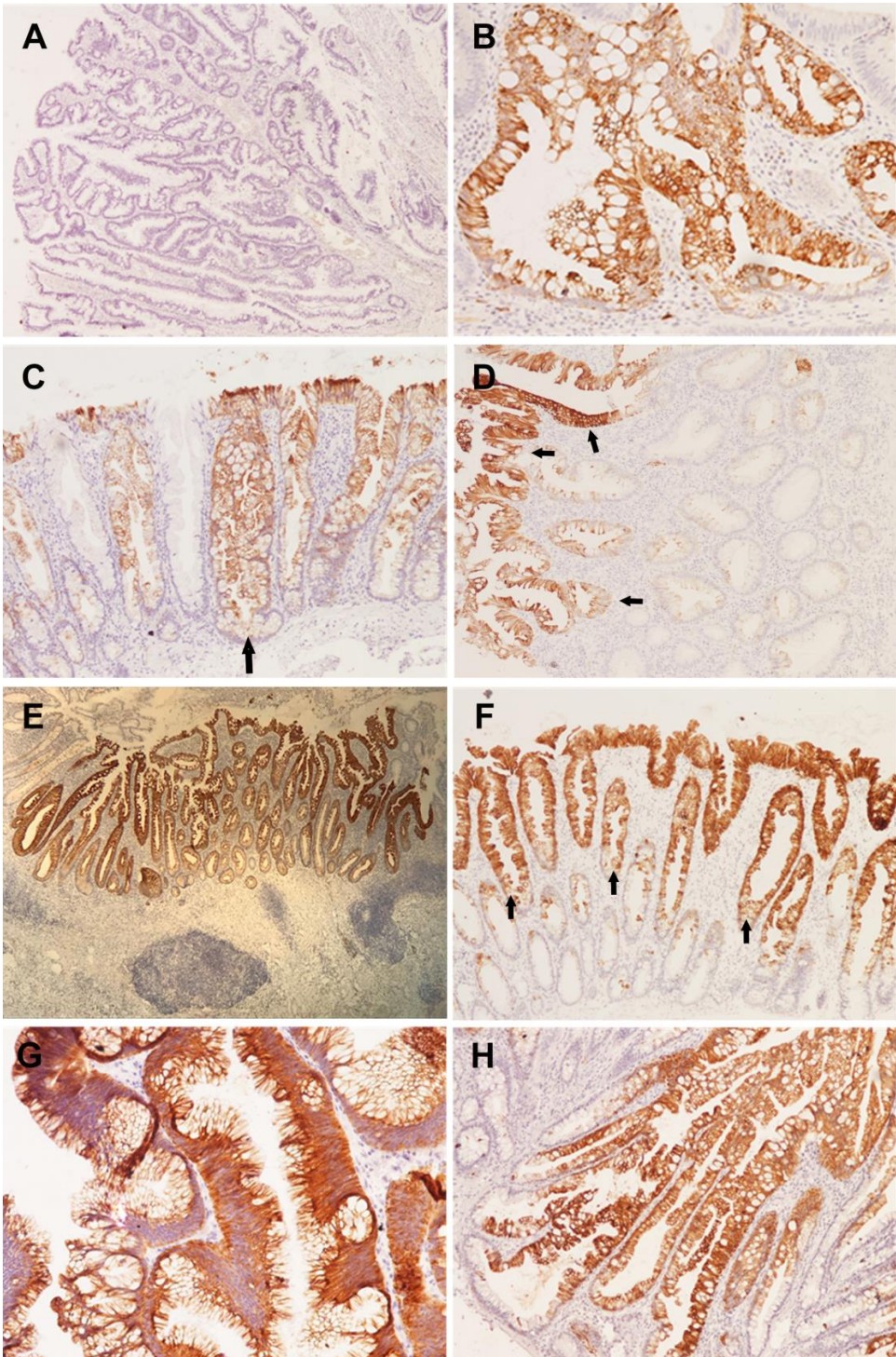

**Figure 2.** (**A**) Absence of CK7 immunoreactivity in TSA (LSAB2 × 40); (**B**) positive expression of CK7 in basal parts of a mixed serrated lesion (LSAB2 × 200); (**C**) CK7 positivity of the entire crypt in the area of pronounced serration and dilatation with horizontal growth in SSL without dysplasia (arrow) (LSAB2 × 100); (**D**) expression of CK7 exclusively in the upper half of the crypt (left) in HP (arrows) (LSAB2 × 100); (**E**) hyperplastic appendiceal polyp presenting as a flat lesion, clearly demarcated by uniform expression of CK7 (LSAB2 × 40); (**F**) hyperplastic polyp in which the upper half of the crypts are clearly demarcated by CK20 positivity (arrows) (LSAB2 × 100); (**G**) complete positivity of superficial epithelium and ectopic crypt foci for CK20 in TSA (LSAB2 × 200); (**H**) CK20 expression dominantly in the upper half of the crypt spreading towards the basal parts, but without involvement of the entire crypt in SSL without dysplasia (LSAB2 × 100).

**Table 2.** Relationship between the serrated lesion category and CK7 expression.

| Microscopic Appearance/ck7 | Negative | Positive Lower Half of the Crypt | Positive Upper Half of the Crypt | Positive Complete Crypt | $\sum$ (n) |
|---|---|---|---|---|---|
| HP | 8 | 0 | 24 * | 2 | 34 |
| SSL without dysplasia | 0 | 0 | 17 | 10 * | 27 |
| SSL with dysplasia | 4 | 0 | 7 | 0 | 11 |
| TSA | 6 * | 0 | 0 | 0 | 6 |
| Mixed serrated lesion | 5 | 0 | 9 | 7 | 21 |
| Unclassified serrated lesion | 2 | 0 | 0 | 0 | 2 |
| Malignantly altered serrated lesion | 6 * | 1 | 0 | 1 | 8 |
| $\sum$ (n) | 19 | 1 | 33 | 10 | 109 |

\* $p < 0.01$.

**Table 3.** Relationship between CK20 positivity in different parts of the crypts and the serrated lesion subtype.

| Microscopic Appearance/ck20 | Negative | positive Lower Half of the Crypt | Positive Upper Half of the Crypt | Positive Complete Crypt | $\sum$ (n) |
|---|---|---|---|---|---|
| HP | 0 | 0 | 31 * | 3 | 34 |
| SSL without dysplasia | 0 | 0 | 24 | 3 | 27 |
| SSL with dysplasia | 0 | 0 | 7 | 4 | 11 |
| TSA | 0 | 0 | 0 | 6 * | 6 |
| Mixed serrated lesion | 0 | 0 | 18 | 3 | 21 |
| Unclassified serrated lesion | 2 * | 0 | 0 | 0 | 2 |
| Malignantly altered serrated lesion | 0 | 0 | 3 | 5 | 8 |
| $\sum$ (n) | 0 | 0 | 52 | 18 | 109 |

\* $p < 0.01$.

Sessile serrated lesions without dysplasia and sessile serrated lesions with dysplasia did not show a statistically significant distribution of CK20 positivity in relation to a certain part of the crypt ($p > 0.01$) (Figure 2H).

## 4. Discussion

"Serrated" lesions are a relatively recently discovered group of colorectal neoplasms characterized by the serrated structure of hyperplastic polyps and cellular atypia of tubular adenomas [15]. The classification and diagnostic criteria have changed in the last two decades [16], but in this analysis, the frequency and representation of individual subtypes according to the literature data have not changed significantly.

Hyperplastic polyps (HP) are the most common serrated lesions, represented in more than 75% of all serrated lesions [17]; in this study, the frequency was twice as low and amounted to 31.2%, which can be explained by the fact that hyperplastic polyps, which occurred concurrently with conventional adenomas and the non-serrated subtype of carcinomas, were not included in the analysis. Likewise, the frequency of HP of 31.2% in this study was higher than the individual frequency of all other serrated lesions, but without statistical significance ($p = 0.12$).

The average age of patients did not show a statistically significant difference between the subtypes of serrated lesions ($p > 0.05$), which was not particularly emphasized by other authors [15,18,19]. In addition, in this study, the highest average age was in patients with TSA (69.8 ± 11.2 years), and the lowest in patients with SSL without dysplasia (61.7 ± 12.9 years). According to Jass, depending on the territorial affiliation of the studied

population, they occur more often in men than in women with a ratio of 2:1 [18], which was also confirmed by the results of this study where the frequency of serrated lesions in men was in fact2.4 times higher than in women (70.6% vs. 29.4%); however, Matsumoto et al. found no significant gender difference in the frequency of serrated adenomas in their research [19].

Serrated appendiceal polyps occur in men and women with almost identical frequency [13]; in this analysis, serrated appendiceal lesions were 2.7 times more common in male patients. Patients with serrated appendiceal lesions are mostly older adults, in their sixties to eighties, although a broad range is present [13].

According to the literature data, they are found along the entire large intestine, with about 11% in the proximal part of the colon including the appendix, which is almost identical to the results of this analysis, where the frequency of serrated lesions in the appendix was 10.1%, but without statistical significance according to other localizations ($p > 0.05$). Otherwise, they are most commonly found in the sigmoid colon and rectum (53.6% in Western countries [15] and 54–67.3% in Japan [19,20]). The incidence of serrated lesions in the sigmoid and rectum of 56% within this study is closer to the frequency reported for Western countries compared to Japan, where the average age of patients with serrated lesions is higher. Considering data from the literature, they have a significant malignant potential, with cancer foci found in 11% of all cases [20], and the results of this analysis have confirmed the frequency of malignantly altered serrated lesions to be 7.34% and statistically significantly more common in the anal canal than in other segments of the colon ($p = 0.025$).

Sessile serrated lesions have a preference for the proximal colon (70–80%), especially those with dysplasia [3], which was not confirmed by the results of this study where the frequency of all sessile serrated lesions in the proximal colon was 39.5%, of which frequency of SSL without dysplasia was 31.6% and SSL with dysplasia was 7.9%, but without a statistically significant difference ($p > 0.05$). On the other hand, SSL without dysplasia showed a statistically significantly higher frequency of occurrence in the ascending and sigmoid part of the colon compared to other serrated lesions of the same localization ($p < 0.05$).

About 70% of traditional serrated adenomas occur in the distal colon and rectum [21], which is in accordance with the results of this study where, in 100% of the cases, they were localized in the sigmoid colon and rectum, but without a statistically significant difference ($p > 0.05$). The reason why all TSAs in this analysis were found in the distal parts of the large intestine lies in the fact that the TSAs, which are a component of mixed serrated lesions and were also present in the proximal segments, were not shown to be broken down according to localization in the large intestine.

Proximal TSAs can be flat, but TSAs are usually polypoidal, broad lesions, with a surface resembling a pinecone or coral surface [21], which in this study was confirmed to be statistically significant only for the semi-sessile growth manner of TSAs ($p = 0.035$), but not for the 'flat' manner of growth, because not a single case of TSA had such a macroscopic aspect to it.

As reported in some papers, serrated lesions of the appendix can form a discrete polyp or circumferentially involve the mucous membrane of the appendix [1]; in this analysis, serrated lesions of the appendix presented in 10.1% of the cases as flat lesions that were inconspicuous macroscopically, but this type of growth was not statistically significantly different from the appearance of flat, discrete forms of serrated lesions at other locations ($p > 0.05$). This macroscopic manner of growth can make it difficult to recognize them even at the microscopic level, and in this research, such cases were solved by using the CK7 marker, which clearly demarcated them (Figure 2E).

According to the findings of some authors, there is an obvious difference in the expression pattern of CK7/CK20 in serrated lesions and conventional adenomas, so that most serrated lesions have a CK7+/CK20+ immunophenotype, in contrast to conventional adenomas that have a CK7−/CK20+ immunophenotype [22]. However, this study particularly highlighted the immunophenotype of CK7 and CK20 in certain categories of serrated

lesions depending on the part of the crypt where the expression is present, which was not specifically examined in the available literature. In addition, the reports of the same authors indicate that the immunohistochemical expression of CK7 can be a marker of the serrated pathway of colorectal carcinogenesis [22].

Accordingly, the serrated appearance of the surface is primarily a consequence of the morphological changes caused by the enlargement and budding of the adenoma, as well as the migration of its epithelium at the crypt level and the creation of elevations that resemble saw teeth [18]; in this study, the serrated morphology was particularly highlighted immunohistochemically using CK7 and CK20 markers in hyperplastic polyps ($p < 0.01$), in sessile serrated lesions without dysplasia using CK7 ($p = 0.005$), in TSA using CK20 ($p < 0.01$), and in sessile serrated lesions with dysplasia, mixed serrated lesions and malignantly altered serrated lesions without statistical significance using CK7 and CK20 ($p > 0.01$). Such results could prove important in subsequent studies, as the first reports revealed a correlation between BRAF mutations, microsatellite status, and CK7 and CK20 expression, as well as decreased CK20 expression in colorectal cancers with microsatellite instability [10].

The claims of certain authors that immunohistochemical application of CK20 stains only superficial epithelial cells, but not ectopic crypt foci in TSA [23], were not confirmed by the findings of this study because CK20 positivity of complete crypts, including the ectopic crypt formations (Figure 2G), was a statistically significant feature of TSA compared to other serrated lesions ($p < 0.01$).

Although the histopathological differences between individual subtypes of serrated lesions are usually sufficient for pathologists to make a definitive diagnosis, the differences between subtypes of serrated lesions with different expressions of cytokeratins and mucin glycoproteins remain a matter of debate [24–27]. In this sense, this study singled out TSA and malignantly altered serrated lesions as lesions that are statistically significantly more often negative for CK7 expression ($p < 0.01$), while CK20 positivity of the complete crypts was shown to be a statistically significant feature of TSA, and CK20 positivity of the upper half of the crypts a statistically significant feature of hyperplastic polyps ($p < 0.01$).

Other published data about this topic of CK7 and CK20 expression in serrated lesions did not point out the pattern of distribution of these markers along the crypts. The novelty of our work is that is does precisely that, helping in the differential diagnosis of serrated lesions, even on scant or poorly oriented endoscopic biopsies.

## 5. Conclusions

Serrated lesions of the large intestine represent a relatively heterogeneous group with overlapping micromorphological characteristics, and analysis of the micromorphological characteristics of serrated lesions with adequately interpreted immunohistochemical results contributes to their differentiation, even on endoscopic biopsy. We believe that differences in the expression patterns of cytokeratin 7 and 20 in different serrated lesions may indicate different pathways of colorectal carcinogenesis and be diagnostically and prognostically useful in future studies on their malignant potential.

**Author Contributions:** Conceptualization, I.I.; methodology, I.I. and B.R.D.; software, P.R.; validation, Ž.M., and M.J.M.; investigation, B.R.D.; data curation, P.R.; writing—original draft preparation, I.I.; writing—review and editing, I.I. and J.C.; supervision, Ž.M and M.J.M.; project administration, J.C. All authors have read and agreed to the published version of the manuscript.

**Funding:** This work was supported by the Ministry of Education, Science and Technological Development, Republic of Serbia under grant number 451-03-68/2022-14/200113.

**Institutional Review Board Statement:** Not applicable.

**Informed Consent Statement:** Not applicable.

**Data Availability Statement:** Data sharing not applicable.

**Conflicts of Interest:** The authors declare no conflict of interest. The funders had no role in the design of the study; in the collection, analyses, or interpretation of data; in the writing of the manuscript; or in the decision to publish the results.

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
