# Peer review of "Significance of Micromorphological Characteristics and Expression of Intermediate Filament Proteins CK7 and CK20 in the Differential Diagnosis of Serrated Lesions of the Colorectum"

_gastroent, doi:10.3390/gastroent14010008_

Round 1

Reviewer 1 Report

Ilic et al presented a manuscript titled: "Significance of micromorphological characteristics and expression of intermediate filament proteins CK7 and CK20 in the differential diagnosis of serrated lesions of the colorectum". Unfortunately there are some major weaknesses in your work.

In the Methods section, there are no inclusion or exlusion criteria. How did you include your patients? Furthermore, there is no ethical comittee approval presented in your study.

Your Statistical analyses subsection of the Methods section is very scarce. What method did you use to determine the normality of distribution? How did you present your quantitative and qualitative varibles? What statistical methods were used for the comparison between groups? What statistical methods did you use to estimate correlation? Unfortunately, you did not mention this imporant information in the results section either. Your tables and figures are missing this elaborations, hence, this is a major weakness of your study. It is impossible to objectively inetepret your outcomes if I'm not sure were the biostatistical methods appropriate.

Furthermore, apart from this major weakness regarding your methodology, this study does not provide any significant scientific novelty. All of these issues are already well-established and investigated.

Investigating through literature, there are several articles regarding this topic. In your study you haven't pointed out the novelty of your work

Author Response

We have inserted new text into manuscript according to reviewer suggestion. We agree about all comments, new version will provide more answers to questions asked.

We have included inclusion and exclusion criteria in the Methods section.

We have described normality test used as well as other statistical tests we used. Not mentioned them was a huge unintentional mistake we made.

We have added at end of Discussion section what we think is the novelty in our work, as proposed by reviewer.

We do not have ethical comittee approval since material used in the study was routinely analysed for diagnostic purposes already. Our ethical comittee does not ask for permission in such case.

Reviewer 2 Report

In the manuscript submitted for review, the Authors described the importance of micromorphological features and expression of the intermediate CK7 and CK20 filament proteins in the differential diagnosis of serrated lesions of the large intestine.

I find the topic interesting and very important nowadays. The structure of the reviewed article is well thought out and clear. In the introduction, the Authors introduced the reader to the subject of the manuscript in depth, and the conclusions they drew from the discussions are clear and legible. My only comment concerns the photos - maybe a positive reaction should be marked in some way (arrow??) in the photo. Especially for novice readers, it would be more readable. But that's just my suggestion. The work is well written and I recommend it for publication.

Author Response

We have inserted the arrows into Figure 2 as recommended by reviewer. The positive reaction is marked by arrows.

Round 2

Reviewer 1 Report

The authors have really significantly improved the quality of their manuscript. The results can now be properly interpreted and the discussion gave an elaborated in-depth analysis of their outcomes.

Author Response

Thank you for your comments. You helped us to make a better article following your guidance.